# Aux-Drop: Handling Haphazard Inputs in Online Learning Using Auxiliary Dropouts

**Rohit Agarwal**                                                                              *agarwal.102497@gmail.com*
*Bio-AI Lab, Department of Computer Science*
*UiT The Arctic University of Norway, Tromsø*

**Deepak Gupta**
*Bio-AI Lab, Department of Computer Science*
*UiT The Arctic University of Norway, Tromsø*

**Alexander Horsch**
*Bio-AI Lab, Department of Computer Science*
*UiT The Arctic University of Norway, Tromsø*

**Dilip K. Prasad**
*Bio-AI Lab, Department of Computer Science*
*UiT The Arctic University of Norway, Tromsø*

**Reviewed on OpenReview:** *https://openreview.net/forum?id=R9CgBkeZ6Z*

## Abstract

Many real-world applications based on online learning produce streaming data that is haphazard in nature, *i.e.*, contains missing features, features becoming obsolete in time, the appearance of new features at later points in time and a lack of clarity on the total number of input features. These challenges make it hard to build a learnable system for such applications, and almost no work exists in deep learning that addresses this issue. In this paper, we present Aux-Drop, an auxiliary dropout regularization strategy for online learning that handles the haphazard input features in an effective manner. Aux-Drop adapts the conventional dropout regularization scheme for the haphazard input feature space ensuring that the final output is minimally impacted by the chaotic appearance of such features. It helps to prevent the co-adaptation of especially the auxiliary and base features, as well as reduces the strong dependence of the output on any of the auxiliary inputs of the model. This helps in better learning for scenarios where certain features disappear in time or when new features are to be modelled. The efficacy of Aux-Drop has been demonstrated through extensive numerical experiments on SOTA benchmarking datasets that include Italy Power Demand, HIGGS, SUSY and multiple UCI datasets. The code is available at `https://github.com/Rohit102497/Aux-Drop`.

## 1 Introduction

Many real-life applications produce streaming data that is difficult to model. Moreover, a lot of existing methods assume that the streaming data has a time-invariant fixed size and the models are trained accordingly (Gama, 2012; Nguyen et al., 2015). However, this is not always true and the dimension of inputs can vary over time. The inputs can have missing data, missing features, obsolete features, sudden features and an unknown number of the total features. We define this here as the *haphazard inputs*. Formally, we define haphazard inputs as streaming data whose dimension varies at every time instance and there is no prior information about data received in the future. The characteristics of haphazard inputs are as follows: (1) *Streaming data* - Here data arrives sequentially and is modelled using online learning techniques. The model

**Table 1:** Comparison of different online deep learning models with respect to the characteristics of the haphazard inputs (C1-C6). We showcase the inability of online deep learning methods in handling haphazard inputs even when other techniques like imputation, extrapolation, priori information and Gaussian noise are employed.

| Characteristics | Aux-Drop | Online Deep Learning Methods like ODL | ODL + Online Data Imputation | ODL + Extrapolation | ODL + Prior Information | ODL + Gaussian Noise |
|---|---|---|---|---|---|---|
| Streaming data (C1) | ✓ | ✓ | ✓ | ✓ | ✓ | ✓ |
| Missing data (C2) | ✓ | × | ✓ | × | ✓ | ✓ |
| Missing features (C3) | ✓ | × | × | × | × | ✓ |
| Obsolete features (C4) | ✓ | × | × | ✓ | × | ✓ |
| Sudden features (C5) | ✓ | × | × | × | × | × |
| Unknown no. of features (C6) | ✓ | × | × | × | × | × |

predicts an output based on the current data instance and then the actual output is revealed. The model gets trained based on the loss from its prediction and the actual output, and this updated model is used for future prediction (Hoi et al., 2021). (2) *Missing data* - The input features can be missing at any time instance. It can be due to data corruption, malfunctioning sensors, faulty equipment, human errors, etc. (Emmanuel et al., 2021). (3) *Missing features* - It is known that a certain feature will arrive but doesn't have any other prior information like its distribution. It is never received at time instance $t = 1$. (4) *Obsolete features* - The input features are received at any point in time but it ceases to exist after some time instances. (5) *Sudden features* - There is no information about the existence of these features when the model is defined. The model might know about this feature suddenly at any point of time after the model deployment. (6) *Unknown number of features*: At the time of model designing, there is no information about the total number of input features and at no point in time this information is available.

Some approaches (Beyazit et al., 2019; He et al., 2019) are available to address the challenges of haphazard inputs but there is a dearth of deep-learning approaches in this field. Moreover, the characterization of the problem discussed here is not defined properly in the previous works, hence we make an attempt to introduce the problem (haphazard inputs) in a more formal way in this paper.

The outlined issues of haphazard inputs can be partly tackled by coupling an online deep-learning framework with some existing approaches such as feature imputation, extrapolation of information and regularization with Gaussian noise, among others. But not all challenges can be handled by any single framework. This is better explained in Table 1 where we list the prominent challenges of online learning with streaming data as well as point out the limitations of the existing approaches. Since the data is streaming, it becomes imperative to apply online learning methods like online deep learning (ODL) (Sahoo et al., 2017) and online gradient descent (OGD) (Sahoo et al., 2017; Cesa-Bianchi et al., 1996), however, it can only handle the streaming aspect of the haphazard input. The online imputation model can be used to impute missing data (C2) and can thus be applied in conjunction with any other online learning method but still, it can't address the other characteristics. Extrapolation can be used to address the case of obsolete features (C4). Prior information on the features can be used to project the missing data (C2). Lastly, there can be a naive way of employing the Gaussian noise wherever the data is not available. This can address missing data (C2), missing features (C3) and obsolete features (C4), however, it can still not cater to the appearance of new features (C5) as well as handle the issue of missing information on the total number of features (C6).

Recently, Agarwal et al. (2020) presented Aux-Net, a deep learning architecture capable of handling the issues outlined above. However, Aux-Net employs a dedicated layer for each auxiliary feature, which results in a very heavy overall network, and this leads to a significant increase in training time for each additional auxiliary feature being modelled. The high time and space complexity of Aux-Net makes it inefficient and not scalable for larger problems/datasets. Here, auxiliary features refer to those features which are not available consistently in time, rather these are subjected to atleast one of the characteristics (C2-C6) as outlined in Table 1. It is assumed that atleast one feature is always available and is known as the base

feature. Let us denote the set of indices of base features received at time $t$ by $\mathbb{B}_t = \mathbb{B}$, then the set of indices of base features received at time $t+1$ is $\mathbb{B}_{t+1} = \mathbb{B}$. Similarly, let us denote the set of indices of auxiliary features received at time $t$ by $\mathbb{A}_t$, then the set of indices of auxiliary features received at time $t+1$ is given by $\mathbb{A}_{t+1} \subseteq [\cup_{i=1}^{t} \mathbb{A}_i] \cup \widetilde{\mathbb{A}}_{t+1}$ where $\widetilde{\mathbb{A}}_{t+1}$ is the set of indices of new auxiliary features received such that $\widetilde{\mathbb{A}}_{t+1} \cap [\cup_{i=1}^{t} \mathbb{A}_i] = \phi$. Here, we follow the assumption that $\mathbb{B} \neq \phi$, *i.e.*, there is atleast one base feature.

In this paper, we present *Aux-Drop*, an auxiliary dropout regularization strategy for an online learning regime that handles the haphazard input features in an accurate as well as efficient manner. Aux-Drop adapts the conventional dropout regularization scheme (Hinton et al., 2012) for the haphazard input feature space ensuring that the final output is minimally impacted by the chaotic appearance of such features. It helps to prevent the co-adaptation of especially the auxiliary and base features, as well as reduces the strong dependence of the output on any of the auxiliary inputs of the model. This helps in better learning for scenarios where certain features disappear in time or when new features are to be modelled. Aux-Drop is simple and lightweight, as well as scalable to even a very large number of auxiliary features. We show the working of our model on the Italy Power Demand dataset (Dau et al., 2019), the widely used benchmarking datasets for online learning such as HIGGS (Baldi et al., 2014) and SUSY (Baldi et al., 2014) and 4 different UCI datasets (Dua & Graff, 2017).

The contributions of this paper can be listed as follows: (1) We formally introduce the problem of haphazard inputs and their characteristics. (2) We propose a dropout-inspired concept called Aux-Drop to handle the haphazard streaming inputs during online learning. It employs selective dropout to drop auxiliary nodes accommodating the haphazard auxiliary features and random dropout to drop other nodes. Together they handle the auxiliary features while preventing co-adaptations of auxiliary and base features. (3) The simplicity of Aux-Drop allows us to couple it with existing deep neural networks with minimal modifications and we demonstrate it through ODL and OGD.

## 2 Related Work

**Online Learning**  Online learning is approached via multiple concepts in the machine learning domain (Gama, 2012; Nguyen et al., 2015). Among the various approaches that exist, some popular methods are k-nearest neighbours (Aggarwal et al., 2006), decision trees (Domingos & Hulten, 2000), support vector machines (Tsang et al., 2007), fuzzy logic (Das et al., 2016; Iyer et al., 2018), bayesian theory (Seidl et al., 2009) and neural networks (Leite et al., 2013). Recently, deep learning approaches with different learning mechanisms (Hoi et al., 2021) are introduced resulting in architectures like online deep learning (ODL) (Sahoo et al., 2017). The ODL has shown tremendous improvement in the learning capability for streaming classification tasks. But all these methods are limited by the assumption of fixed input features.

**Haphazard Inputs**  Different versions and subsets of haphazard inputs are present in the literature. Incremental learning approaches like ensemble methods (Polikar, 2012), Learn++ algorithms (Polikar et al., 2001; Mohammed et al., 2006), Learn++.MF (Polikar et al., 2010) and Learn++.NSE (Elwell & Polikar, 2011) handles only missing feature problems and is too expensive in terms of training and storage requirements. Zhou (2022) presents *open-environment* machine learning which includes emerging new classes (Parmar et al., 2021), incremental/decremental features (Hou et al., 2021), changing data distribution (Sehwag et al., 2019) and varied learning objectives (Ding & Zhou, 2018). Online learning with streaming features (OLSF) algorithm (Zhang et al., 2016) handles the *trapezoidal data streams* where both data volume and feature space increase over time. Hou et al. (2017) introduced the problem of *feature evolvable streams (FESL)* where the set of features changes after a regular time period. Zhang et al. (2020) proposed evolving discrepancy minimization (EDM) for data with *evolving feature space and data distribution*. Hou et al. (2021) tries to solve an interesting problem where there is an overlap between old and new features by introducing an incomplete overlapping period. Evolving metric learning (EML) (Dong et al., 2021) handles the *incremental and decremental features*. All the above methods solve specific problems in online learning and are only a subpart of the haphazard inputs. Online learning from varying features (OLVF) (Beyazit et al., 2019) and online learning with capricious data streams (OCDS) (He et al., 2019) can model haphazard inputs but both of them are non-deep-learning approaches and are only tested in small datasets. OCDS trains a learner based on a universal feature space that includes the features appearing at each iteration. It reconstructs the

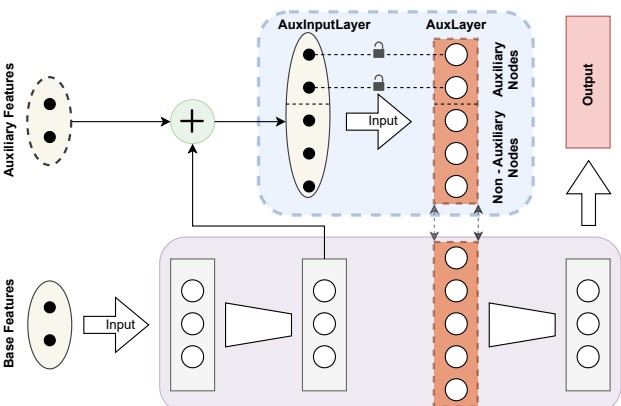

**Figure 1:** (Best viewed in color) The Aux-Drop architecture. The purple-colored box represents any online learning-based deep learning approach. The trapezoid denotes zero or more fully connected layers. Both the brown boxes are the same (represented by double-headed arrows) and are known as AuxLayer. The AuxInputLayer is the concatenation of the hidden features from the layer previous to the AuxLayer and the auxiliary features. The AuxInputLayer and the AuxLayer are fully connected but depending on the unavailability of auxiliary features, the corresponding nodes are dropped (termed Auxiliary Nodes). The unlocked lock denotes an inherent one-to-one connection between the auxiliary features and the auxiliary nodes.

unobserved instances from observable instances by capturing the relatedness using a graph. Thus, OCDS is based on the dependency between features. Online learning from varying features (OLVF) (Beyazit et al., 2019) tries to handle the varying features by projecting the instance and classifier at any time $t$ into a shared feature subspace. It learns to classify the feature spaces and the instances from feature spaces simultaneously. The transformation in different feature spaces leads to a loss of information resulting in poorer performance.

**Dropout**  Dropout was proposed by Hinton et al. (2012) to prevent co-adaptations between features. Since then dropout has been used in different settings to handle various problems. Spatial dropout (Tompson et al., 2015) removes adjacent pixels in convolutional feature maps to reduce strong spatial correlation in the feature map activations. Poernomo & Kang (2018) proposed biased dropout and crossmap dropout for better generalization of the model. Biased dropout divides the nodes in a layer into two groups, the one with higher activation has a lower dropout rate and the one with lower activation has a higher dropout rate. Crossmap drops or retains the nodes simultaneously in equivalent positions of each feature map to break the correlation between adjacent maps. Stochastic Activation Pruning (SAP) (Dhillon et al., 2018) preserves the nodes with higher activation magnitude and normalizes the retained node to preserve the dynamic range of activations in each layer. Guided dropout (Keshari et al., 2019) drops the nodes with higher strength in order to improve the performance of low-strength nodes. Group-wise dropout (Ke et al., 2020) adjusts the dropout probabilities adaptively using the feature density by analyzing the number of linearly uncorrelated deep features gathered over equally spaced grids in low-dimension feature space. All the discussed dropout strategy is employed to better generalize the model and prevent overfitting but none of these strategies can be used to handle haphazard inputs. Aux-Drop drops the nodes in the hidden layer based on the unavailability of auxiliary features and dropout rate.

## 3 Method

### 3.1 Aux-Drop

The core of Aux-Drop lies in utilizing the concept of dropout to accommodate the ever-changing characteristics of haphazard inputs. Dropout drops the nodes randomly from a hidden layer whereas we employ selective dropout along with the random dropout. The proposed Aux-Drop concept handles the base features and auxiliary features synchronously. The input features which are always available are termed base features and the input features which are haphazard inputs are known as auxiliary features. The Aux-Drop concept can be applied in any deep learning-based model capable of handling streaming data. A conventional online deep learning model has one input layer which is connected to the first hidden layer and all the input features are passed via this input layer. But in the Aux-Drop setup, we created a division in the passing of input features to the model such that it can utilize all the information from the base features and increment the model learning from the haphazardly available auxiliary features. The base features are directly passed to the deep learning model (the purple color box in Figure 1). A hidden layer of this model is designated as an AuxLayer and is represented by the dashed brown rectangular box in Figure 1. The hidden features from

the layer previous to the AuxLayer are concatenated with the incoming auxiliary features and are known as the AuxInputLayer. The input to the AuxLayer is the AuxInputLayer and is fully connected. Based on the number of different auxiliary features received, a pool of auxiliary nodes is chosen from the AuxLayer such that there is a correspondence between an auxiliary feature and a specific auxiliary node. Whenever an auxiliary feature is not available, the corresponding auxiliary node is dropped from the AuxLayer. This creates an inherent one-to-one connection between the auxiliary features and the auxiliary nodes. The rest of the nodes in the AuxLayer are termed the Non-Auxiliary nodes. The diagram of the Aux-Drop concept is presented in Figure 1.

Dropout is applied only in the AuxLayer. The nodes to be dropped include all those nodes from the pool of auxiliary nodes whose corresponding auxiliary features are unavailable, forming the group of selective dropout nodes. The rest of the dropout nodes are chosen randomly from the remaining nodes of AuxLayer. Dropout (Hinton et al., 2012) was proposed to prevent complex co-adaptations on the training data. We exploit this property of dropout in handling haphazard inputs. Instead of randomly dropping the nodes, we define some auxiliary nodes that certainly need to be dropped. Aux-Drop makes auxiliary features contribute to the deep learning model even when some of the other features are not available making it independent. This prevented complex co-adaptations in which an auxiliary feature is only helpful in the context of several other auxiliary features.

## 3.2 Mathematical Formulation

**Problem Statement** The problem statement is defined as finding a mapping $f : X \rightarrow Y$, where $X, Y$ is streaming data, such that $X, Y = \{(X_1, Y_1), ..., (X_T, Y_T)\}$. The capital letter variables in italics denote a vector here. The input $X$ consists of base features $(X^B)$ and auxiliary features $(X^A)$ and can be represented as $X = \{X^B, X^A\}$. We define $n_B$ as the number of base features and $n_A^t$ as the number of auxiliary features received at time $t$. For convenience, we define $n_A^{max}$ as the maximum number of auxiliary features. Note that, we do not need the information about $n_A^{max}$ at any point in time in the model. The input feature at time instance $t$ is given by $X_t = \{X_t^B, X_t^A\}$ where $X_t^B$ and $X_t^A$ are the base features and the auxiliary features at time $t$, respectively. Let us denote an input feature by $x$, then the base features at any time $t$ is given by $X_t^B = \{x_{j,t}^B\}_{\forall j \in \mathbb{B}}$, where $\mathbb{B}$ is the set of indices of base features such that $\mathbb{B} = \{1, ..., b, ..., n_B\}$ and $b$ is the index of $b^{\text{th}}$ base feature. Similarly, the auxiliary features at any time $t$ is given by $X_t^A = \{x_{j,t}^A\}_{\forall j \in \mathbb{A}_t}$, where $\mathbb{A}_t$ is the set of indices of auxiliary features at time $t$ such that $\mathbb{A}_t \subseteq \mathbb{A} = \{1, ..., a, ..., n_A^{max}\}$ and $a$ is the index of $a^{\text{th}}$ auxiliary feature. The output $Y \in \mathbb{R}^c$, where $c$ is the total number of classes. Since the problem is based on online learning, at any time $t$, we have access to only the input features $X_t$ and once the model is trained on $X_t$ and a prediction is made, we get the output labels $Y_t$.

**AuxLayer** AuxLayer handles the haphazard auxiliary inputs by employing the dropout. Any $i^{\text{th}}$ layer of the model is mathematically given by $\mathbb{L}_i = \{W_i; S_i; \mathbb{M}_i\}$, where $W_i$, $S_i$ and $\mathbb{M}_i$ denotes the weights connection between the nodes of $(i-1)^{\text{th}}$ and $i^{\text{th}}$ hidden layer, the bias of the $i^{\text{th}}$ layer nodes, and the set of nodes in the $i^{\text{th}}$ layer respectively. If the layer is the 1st layer or the AuxLayer then for $W_i$, the $(i-1)^{\text{th}}$ layer would be the base features or the AuxLayerInput, respectively (see Figure 1). Thus, if $z^{\text{th}}$ hidden layer is chosen as the AuxLayer, then AuxLayer can be mathematically given by $\mathbb{L}_z = \{W_z; S_z; \mathbb{M}_z\}$, where $\mathbb{M}_z$ consists of auxiliary nodes $(\mathbb{M}_z^{\mathbb{A}})$ and non-auxiliary nodes $(\mathbb{M}_z^{\bar{\mathbb{A}}})$. For each auxiliary feature, there is an auxiliary node, i.e., $|\mathbb{M}_z^{\mathbb{A}}| = |\mathbb{A}|$, where $|\cdot|$ represents the cardinality of a set. The set of auxiliary nodes depends upon the number of new auxiliary features received at any time $t$. Thus, whenever a new auxiliary feature arrives, we introduce a new node with a full connection with AuxInputLayer and an inherent one-to-one connection with the auxiliary feature (represented by the unlocked lock in Figure 1) in the AuxLayer and include it in the set of auxiliary nodes. For simplification, from here on, we will consider $n_A^{max}$ as the maximum number of auxiliary features, and thus $|\mathbb{M}_z^{\mathbb{A}}| = n_A^{max}$. Thus, the number of nodes in the set of non-auxiliary nodes is given by $|\mathbb{M}_z^{\bar{\mathbb{A}}}| = |\mathbb{M}_z| - n_A^{max}$. The AuxInputLayer is the input to the AuxLayer and at time $t$ and it is given by

$$I_t^A = \{X_t^A, H_{z-1,t}\} \tag{1}$$

where $H_{z-1,t}$ is the output of the $(z-1)^{\text{th}}$ layer at time $t$.

---

**Algorithm 1** Aux-Drop algorithm

---

**Require:** A deep learning-based online learning model $OL$, dropout $d$, $z$ as the AuxLayer
    Create Aux-Drop(OL) from $OL$ as done in Figure 1
    **while** time $t$ **do**
        Receive $X_t^A, X_t^B$
        Pass $X_t^B$ to Aux-Drop(OL) and get the hidden features $H_{z-1,t}$
        Get $I_t^A$ by eq. 1
        Get $\mathbb{M}_{z,t}^{\mathbb{D}}$ by eq. 4
        Get $W_{z,t}$, $S_{z,t}$ by freezing weights, bias affected by unavailable auxiliary features, and dropped nodes
        Create $\mathbb{L}_{z,t}$ by eq. 5
        Get the prediction $\hat{Y}_t$ of the model Aux-Drop(OL)
        Receive the actual label $Y_t$
        Compute the loss from $Y_t$ and $\hat{Y}_t$
        Update the weights and biases of Aux-Drop(OL) based on the computed loss
    **end while**

---

**Auxiliary Dropout**   Let the dropout value be $d$, then the number of nodes to be dropped is given by $|\mathbb{M}_z| \cdot d$, and the set of dropout nodes is represented by $\mathbb{M}_z^{\mathbb{D}}$. We always choose the value of $d$ sufficiently large such that the number of dropout nodes is always greater than the number of auxiliary nodes. The auxiliary dropout component consists of selective dropout on the auxiliary nodes based on the unavailable auxiliary features and random dropout on the leftover nodes from the AuxLayer. The selective dropout and random dropouts are represented by $\mathbb{M}_z^{\mathbb{D}_s}$ and $\mathbb{M}_z^{\mathbb{D}_r}$, respectively. The selective dropout includes all those nodes from the auxiliary nodes whose corresponding auxiliary features are unavailable and is given by

$$\mathbb{M}_z^{\mathbb{D}_s} = \mathbb{M}_z^{\mathbb{A}} - \mathbb{M}_z^{\mathbb{A}_t} \tag{2}$$

where $\mathbb{M}_z^{\mathbb{A}_t}$ represents the set of auxiliary nodes whose corresponding auxiliary features are available and $\mathbb{U} - \mathbb{V}$ represents the relative complement, i.e., elements that belong to $\mathbb{U}$ and not to $\mathbb{V}$. Consider $p_i$ as the probability of dropout of each $i^{\text{th}}$ node in the AuxLayer. Then because of the selective dropout, $p_j = 1 \ \forall j \in \mathbb{M}_z^{\mathbb{D}_s}$. Random dropout is achieved using the Bernoulli distribution. Our problem statement is a closer match with the Bernoulli distribution because it is a discrete probability distribution that models the probability of a binary outcome for a single trial which is the basis of selective dropout. Other graphical models like the Random Markov field will fit our problem statement only if we take the case when the set of auxiliary features probability can be modelled as a joint distribution like in the computer vision as there is high dependence between adjacent pixels. $|\mathbb{M}_z| \cdot d - |\mathbb{M}_z^{\mathbb{D}_s}|$ nodes are dropped randomly from the leftover nodes of AuxLayer ($\mathbb{M}_z - \mathbb{M}_z^{\mathbb{D}_s}$). Thus the dropout probability ($p_k$) of the leftover nodes and the random dropouts ($\mathbb{M}_z^{\mathbb{D}_r}$) are given by

$$p_k = \frac{|\mathbb{M}_z| \cdot d - |\mathbb{M}_z^{\mathbb{D}_s}|}{|\mathbb{M}_z - \mathbb{M}_z^{\mathbb{D}_s}|} \qquad \forall k \in \mathbb{M}_z - \mathbb{M}_z^{\mathbb{D}_s}$$
$$\mathbb{M}_z^{\mathbb{D}_r} = \{ \ k \ | \ k \in \mathbb{M}_z - \mathbb{M}_z^{\mathbb{D}_s} \wedge m = 1, m \sim Bernoulli(p_k)\} \tag{3}$$

Thus, the set of nodes dropped from the AuxLayer is given by

$$\mathbb{M}_z^{\mathbb{D}} = \mathbb{M}_z^{\mathbb{D}_s} + \mathbb{M}_z^{\mathbb{D}_r} \tag{4}$$

**Algorithm**   Here, we explain the working of the Aux-Drop. We choose a deep learning-based model capable of handling streaming data and name it $OL$. A dropout value $d$ is set and a hidden layer $z$ is chosen as the AuxLayer. We modify OL as done in Figure 1 and term it Aux-Drop(OL). At the time $t$, we receive the base features $X_t^B$ and the auxiliary features $X_t^A$. The base features are passed to Aux-Drop(OL). We compute the hidden features of all the hidden layers before the AuxLayer. Based on $H_{z-1,t}$, the AuxInputLayer ($I_t^A$) is constructed using eq. 1. Now, we have to create the AuxLayer ($\mathbb{L}_{z,t}$) based on the auxiliary features ($X_t^A$)

received at time $t$. $\mathbb{M}_{z,t}^{\mathbb{D}}$ nodes are determined using eq. 4 and are dropped from the AuxLayer. All the weight connections and bias are frozen which are affected by the unavailable auxiliary features and dropped nodes. Thus, the weights and the bias to the AuxLayer at time $t$ are given by $W_{z,t}$ and $S_{z,t}$, respectively. The AuxLayer is given by

$$\mathbb{L}_{z,t} = \{W_{z,t}; S_{z,t}; \mathbb{M}_z - \mathbb{M}_{z,t}^{\mathbb{D}}\} \tag{5}$$

The AuxInputLayer is passed to the AuxLayer and the successive hidden layers computation is done giving a final prediction $\hat{Y}_t$. Finally, the ground truth $Y_t$ is revealed and the loss is computed between $Y_t$ and $\hat{Y}_t$. The weights and biases of the Aux-Drop(OL) are then updated based on this loss. The algorithm of the Aux-Drop is presented in Algorithm 1.

### 3.3  Handling Haphazard Inputs

The auxiliary features are haphazard inputs that exhibit all the six characteristics presented in Table 1. We present a situation with all the characteristics of haphazard inputs and the changes in the Aux-Drop architecture with respect to different characteristics. Consider there are two output features from the hidden layer previous to the AuxLayer. At the time $t-1$, two auxiliary features are available. We present only the upper half of the model which handles the auxiliary features. The architecture presented in Figure 2 (a) represents the model connection after instance $t-1$. The dropout value is set as 0.7. (1) *An Auxiliary Feature is Missing*: Figure 2 (b) presents the change in the architecture when an auxiliary feature is missing. The inherent one-to-one corresponding node is dropped and all the connections that come with it. Also, all the connections from these auxiliary features to all the other nodes are also frozen. Since the dropout value is 0.7, two nodes need to be dropped. One node is randomly chosen from the remaining 3 nodes and is dropped. (2) *Missing data arrives*: The auxiliary feature missing in the previous case (time $t$) arrives at instance $t+1$. Thus, all the auxiliary features arrive. Two nodes are randomly chosen from all the nodes in the auxiliary layer and dropped along with all its connections as shown in Figure 2 (c). (3) *Obsolete features*: At time $t+2$, an auxiliary feature becomes obsolete. Note that the model at any point doesn't know that this feature is obsolete. Thus for this situation, the change in architecture is the same as the missing auxiliary feature. The change is shown in Figure 2 (d). (4) *Sudden features*: At time $t+3$, a sudden auxiliary feature with no prior information arrives. To handle this, a new auxiliary node and all the connections with it are introduced (shown in the black rectangular box in Figure 2 (e)). The connections from this auxiliary feature to all the nodes in AuxLayer are also introduced. The feature that became obsolete at time $t=2$ will not arrive so the corresponding auxiliary node is dropped. Two more nodes are randomly selected to drop. The architectural change is present in Figure 2 (e). (5) *Missing feature arrives*: At time $t+4$, a missing feature arrives whose prior information is unknown. The only attribute known about this feature is that it will arrive. To handle this, either an auxiliary node can be created and assume that this feature is unavailable till it arrives or it can be considered as a sudden feature. In either case, the computation overhead is almost negligible and the model performance will not change. Hence, we can consider it as a sudden feature and the architectural change is the same as Figure 2 (e).

### 3.4  Discussion

**Independent of the Maximum Number of Auxiliary Features**  For simplification of the Aux-Drop explanation, we set a value $n_A^{max}$ as the maximum number of auxiliary features available to the model. As shown in the previous subsection 3.3, sudden features can be easily handled by the model and hence the value of $n_A^{max}$ is not needed.

**Invariant to the Architecture**  Aux-Drop can be applied to any deep-learning architecture capable of handling streaming data. We apply Aux-Drop on the OGD and ODL framework and represent it as Aux-Drop(OGD) and Aux-Drop(ODL) respectively. Here, we demonstrate the change in the architecture of OGD and ODL framework when Aux-Drop is applied to it (see Figure 3). ***Aux-Drop(OGD)***: OGD is a simple deep-learning neural network with multiple hidden layers trained using stochastic gradient descent (Ketkar & Ketkar, 2017). For simplification, we present a 4-layer OGD (top figure in 3(a)). To apply Aux-Drop on OGD, let's say, we first choose the 3rd hidden layer as the AuxLayer and fix the dropout value as 0.5. The hidden activation from the 2nd hidden layer is concatenated with the incoming auxiliary features giving an

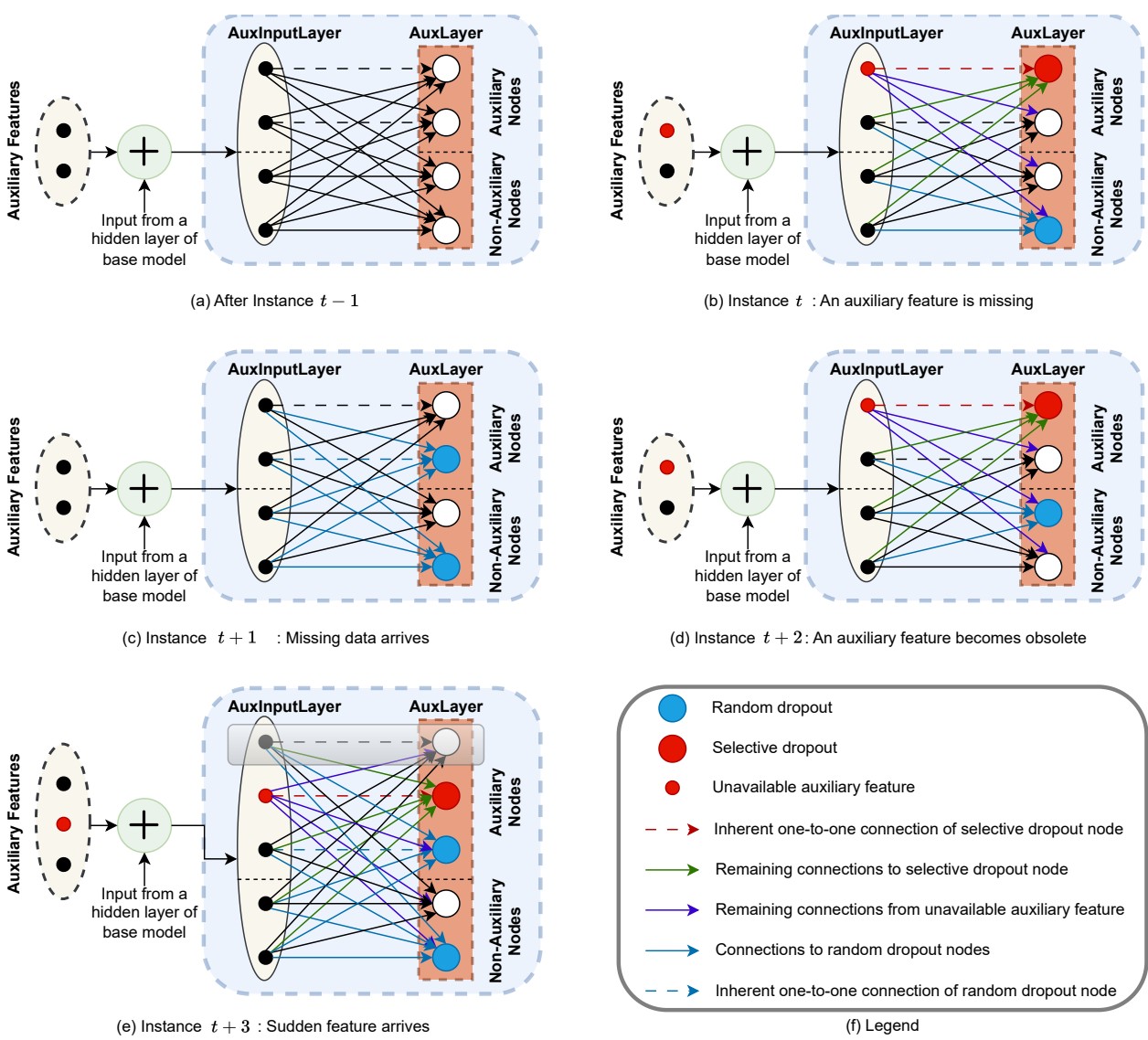

**Figure 2:** (Diagram best viewed in color) We present the changes in the connection between AuxInputLayer and AuxLayer with respect to different characteristics of haphazard inputs. *Different color meaning:* Here the blue circle denotes a random dropout node, and the red circle in the AuxLayer and AuxInputLayer denotes a selective dropout node and unavailable auxiliary feature respectively. The colored arrow follows a hierarchical way of removing connections. First, the inherent one-to-one connection of the selective dropout node is frozen and it is denoted by the dashed red arrow. Next, all the other connections to the selective dropout node are frozen and it is represented by the green arrows. Then all the remaining connections from the unavailable auxiliary features are frozen and are shown by the purple arrows. Next, all the remaining connections to the random dropout node are frozen and are denoted by the blue solid arrows. Finally, the inherent one-to-one connection to the random dropout node is shown by the blue dashed arrow. This is more clear from the legend present in Figure 2 (f). *In Figure 2 (e):* The black rectangular box depicts the arrival of a new auxiliary feature and the introduction of an auxiliary node with all the relevant connections in the AuxLayer. Moreover, all the connections from the new auxiliary feature to all the nodes in the AuxLayer are also introduced.

AuxInputLayer. All the connections between the $2^{\text{nd}}$ and $3^{\text{rd}}$ hidden layer is removed. The $3^{\text{rd}}$ hidden layer is made flexible by increasing the number of nodes and providing it the capability of adding additional nodes whenever required based on the new auxiliary features. The nodes in the AuxLayer are grouped into two parts: auxiliary nodes and non-auxiliary nodes such that the number of nodes in auxiliary nodes is equal

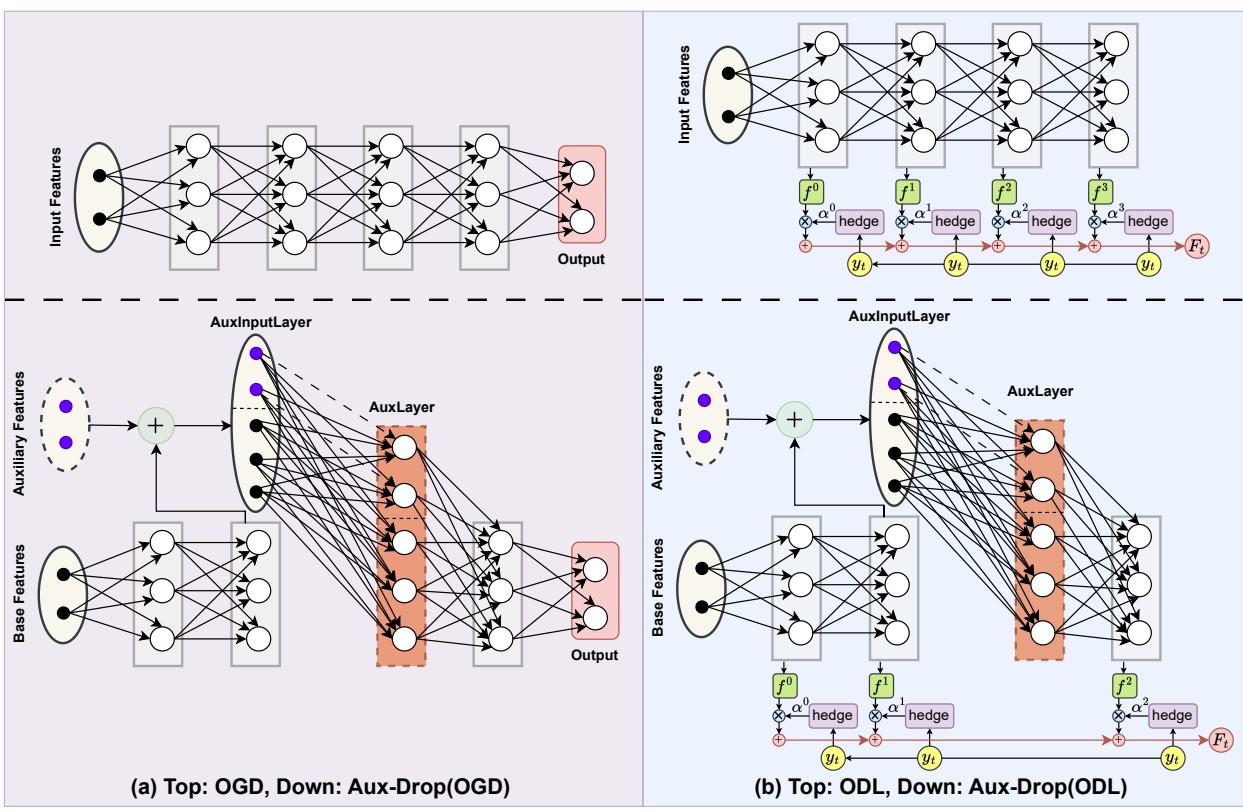

**Figure 3:** (Diagram best viewed in color) Two online learning methods (OGD and ODL) and their corresponding Aux-Drop architecture (Aux-Drop(OGD) and Aux-Drop(ODL)) are shown here. (a) Top: OGD architecture, Down: Aux-Drop(OGD) architecture (b) Top: ODL architecture, Down: Aux-Drop(ODL) architecture.

to the known number of auxiliary features. A full connection is introduced between the AuxInputLayer and AuxLayer such that there is a one-to-one connection between the auxiliary features and the auxiliary nodes. All the required connections (because of the new nodes added in the AuxLayer) between the AuxLayer and the $4^{\text{th}}$ hidden layer are also introduced. This gives us the Aux-Drop(OGD) architecture (see down figure in 3(a)). Selective and random dropout is applied on the AuxLayer and necessary changes are done based on the auxiliary features received at any time instance $t$ as discussed in subsection 3.3. ***Aux-Drop(ODL)***: A snippet of ODL with 4 hidden layers is shown in Figure 3(b) (the top figure). The difference between ODL and OGD is that the output prediction in the OGD architecture is based on the last hidden layer whereas the final prediction in the ODL is based on the weighted output from each hidden layer. The weightage ($\alpha$) of the prediction from each hidden layer is adapted using the Hedge algorithm (Freund & Schapire, 1997). Detailed information about the ODL architecture can be found in (Sahoo et al., 2017). To create Aux-Drop(ODL) (see down figure in 3(b)), we follow a similar process as done for Aux-Drop(OGD). The only difference is that the prediction from AuxLayer is removed since the number of nodes in AuxLayer is always different.

**Haphazard Inputs and Aux-Drop vs Other Works** It is to be noted that haphazard inputs are different than adversarial inputs (Goodfellow et al., 2018) since haphazard inputs are not modified/changed (more discussion in Appendix A). Semi-supervised techniques like MARK (Patil et al., 2022) attempt to fill in the blanks by generating an unavailable pattern for anomaly detection but it is orthogonal to haphazard inputs problem (see Appendix B). One can envision an analogy between progressive networks (Rusu et al., 2016) and Aux-Drop but progressive networks cannot deal with haphazard inputs because of storage and computational inefficiency (see Appendix C). Domain adaptation (Farahani et al., 2021) assumes that all the

**Table 2:** The number of instances and features of all the datasets.

| Dataset | # Instances | # Features |
|---|---|---|
| german | 1000 | 24 |
| Italy Power Demand | 1096 | 24 |
| svmguide3 | 1243 | 21 |
| magic04 | 19020 | 10 |
| a8a | 32561 | 123 |
| SUSY | 1M | 8 |
| HIGGS | 1M | 21 |

features are base features (see Appendix D). Open set problems (Scheirer et al., 2012) deal with an unknown number of classes whereas Aux-Drop deals with an unknown number of input features (see Appendix E).

**Assumption of Base Features**   Aux-Drop assumes that there is atleast one base feature, i.e., atleast one feature is always available. However, the Aux-Drop model can be adapted in multiple ways to remove the assumption of base features as follows: (1) *The naive way to handle this would be to just assume one auxiliary feature as a base feature and impute it. This would induce some bias but still, it would be for just one feature.* (2) *Choosing the position of AuxLayer as the first layer, the Aux-Drop model can be adapted to handle haphazard inputs without any base feature.* But we attempt to propose a generalized concept of Aux-Drop where the position of AuxLayer is a hyperparameter and can be chosen based on the dataset and application. Hence, the assumption of the base feature is adding a better value proposition.

## 4    Experiments

**Datasets** We consider the Italy Power Demand dataset[1] (Dau et al., 2019), HIGGS[2] (Baldi et al., 2014), SUSY[3] (Baldi et al., 2014) and 4 different UCI datasets (german[4], svmguide3[5] (Chang & Lin, 2011), magic04[6], a8a[7]) (Dua & Graff, 2017). The number of instances and the features of each dataset are listed in Table 2. (1) *Italy Power Demand* - It contains the electrical power demand from Italy for twelve months. The associated task is a binary classification task to determine the month of each instance, i.e. distinguish days from Oct to March (inclusive) from April to September. (2) *HIGGS* - It is synthetic data produced using Monte Carlo simulations. It has 28 features out of which the first 21 features are kinematic properties measured by the particle detectors in the accelerator and the last 7 features are the functions of the first 21 features. We use only the first 21 features because it is proved experimentally in the paper (Baldi et al., 2014) that the last 7 features don't contribute to the performance of a deep learning model. The binary classification task is to distinguish between a signal process (1) where new theoretical Higgs bosons (HIGGS) are produced, and a background process (0) with identical decay products but distinct kinematic features. (3) *SUSY* - It is also synthetic data produced using Monte Carlo simulations. The first 8 features are kinematic properties measured by the particle detectors in the accelerator. The last 10 features are functions of the first 8 features. Similar to HIGGS, we only use the first 8 features as it is proved experimentally by Baldi et al. (2014) that the last 10 features do not contribute to training a deep learning model. The associated binary classification task is to distinguish between a signal process (1) where new super-symmetric particles (SUSY) are produced and a background process (0) with the same detectable particles. (4) *german* - It contains the financial information of the customer to classify each customer as having good or bad credit risks. The numerical form of data is considered which has 24 features. (5) *svmguide3* - It is a synthetic dataset of a practical svm guide with a binary classification task containing 1243 instances and 21 features. (6) *magic04* - It is simulated data for the registration of high

---

[1] https://www.cs.ucr.edu/~eamonn/time_series_data_2018/

[2] https://archive.ics.uci.edu/ml/datasets/HIGGS

[3] https://archive.ics.uci.edu/ml/datasets/SUSY

[4] https://archive.ics.uci.edu/ml/datasets/statlog+(german+credit+data)

[5] https://www.csie.ntu.edu.tw/~cjlin/libsvmtools/datasets/binary.html

[6] https://archive.ics.uci.edu/ml/datasets/magic+gamma+telescope

[7] https://archive.ics.uci.edu/ml/datasets/adult

| Probability | Aux-Net | Aux-Drop(ODL) |
|---|---|---|
| .50 | 0.6975 | **0.6031±0.0081** |
| .60 | 0.6831 | **0.5839±0.0111** |
| .70 | 0.6788 | **0.5497±0.0082** |
| .80 | 0.6130 | **0.5321±0.0071** |
| .90 | 0.5456 | **0.5149±0.0119** |
| .95 | 0.5168 | **0.5013±0.0108** |
| .99 | 0.5165 | **0.4788±0.0101** |

**Table 3:** The table contains the average loss on the Italy Power Demand dataset. The first 12 features are base features and the last 12 features are auxiliary features. The availability of each auxiliary feature is varied by a uniform distribution of probability $p$. The value of $p$ ranges from .50 to .99. The average loss of Aux-Net for each $p$ is reported from the Aux-Net paper (Agarwal et al., 2020). Aux-Drop(ODL) is run 20 times and the mean ± standard deviation of the average loss is reported here.

energy gamma particles in an atmospheric Cherenkov telescope. Monte Carlo is used for the simulation. The binary task is to distinguish if a shower image is caused by primary gammas (signal-1) or cosmic rays in the upper atmosphere (background-0). (7) *a8a* - It is census data from 1994 to predict whether the income exceeds \$50k/yr. The pre-processed data to get numerical values has 123 features. The training set of 32561 instances is used for experiments in this paper.

**Motivation of the Chosen Datasets** The motivation to choose these datasets is two-fold: (1) The first is to see how Aux-Drop performs on small (german, Italy Power Demand, svmguide3), medium (magic04, a8a) and large (SUSY, HIGGS) datasets. (2) The number of features in the dataset ranges from 8 (SUSY) to 123 (a8a). This helps in demonstrating the flexibility of Aux-Drop in the varying number of auxiliary features.

**Choice of Deep Learning Model** We use the online deep learning (ODL) model for a few reasons: (a) ODL has shown better performance in the online learning domain and can handle big datasets very efficiently, (b) The only deep learning method available for haphazard inputs (Aux-Net) also use ODL as their base model, hence it gives a fair comparison. We also apply Aux-Drop on the online gradient descent (OGD) framework. We represent them as Aux-Drop(ODL) and Aux-Drop(OGD) respectively and collectively as Aux-Drop.

**Comparison Models** We evaluate the performance of Aux-Drop empirically in multiple scenarios. We compare Aux-Drop(ODL) with Aux-Net (Agarwal et al., 2020) since it is a deep-learning method with ODL base, capable of handling haphazard inputs. We also report Aux-Drop(ODL) and Aux-Drop(OGD) on 4 UCI datasets and compare them with OLVF (Beyazit et al., 2019) and OLSF Zhang et al. (2016). Since most of the datasets used by previous methods were small, we also consider two big datasets to test the effective application and feasibility of Aux-Drop(ODL) and compare it with ODL. In all the scenarios, the instances are provided one by one to the model, and the training and testing are performed in a single pass. For each specific case, the dataset is designed suitably.

## 4.1 Comparison with Aux-Net

The current literature has only one deep learning model, Aux-Net (Agarwal et al., 2020) based on ODL that can handle the situation we present. It is only applied to the Italy Power Demand dataset which is a very small dataset. Nevertheless, we compare our model with Aux-Net and prepare the data similar to the Aux-Net paper. We considered the first 12 features of the Italy Power Demand dataset as the base features and the last 12 features as the auxiliary features. We varied the availability of each auxiliary input independently by a uniform distribution of probability $p$. In comparison with Aux-Drop, Aux-Net is a very heavy model. Let the number of parameters from the base deep learning architecture be $P_B$. Then the number of parameters for Aux-Drop and Aux-Net is given by $P_D$ (eq. 6) and $P_N$ (eq. 7), respectively.

$$P_D = P_B + N_{A_{max}} M_{l_{aux}} \tag{6}$$

$$P_N = P_B + N_{A_{max}} N_H + N_{A_{max}} N_H M_{l_{aux}} \tag{7}$$

where $N_H$ is the number of nodes in the hidden layer. Therefore, the number of parameters in Aux-Net is $N_{A_{max}} * N_H * M_{l_{aux}}$ more than the number of parameters in Aux-Drop since Aux-Net dedicates a layer

**Table 4:** Comparison with OLVF on various datasets. Here all the errors reported for OLVF are on $p = 0.75$ (*i.e.* $Rem = 0.25$) and the value is taken from its original paper. Thus, we adjust the probability value ($p$) for haphazard features for Aux-Drop accordingly to match the amount of missingness of OLVF. The error is reported as the mean $\pm$ standard deviation of the 20 experiments performed with random seeds.

| Dataset | OLVF | Aux-Drop(ODL) | Aux-Drop(OGD) | $p$ |
|---------|------|---------------|---------------|-----|
| german | 333.4±9.7 | **300.4±4.4** | 312.8±19.3 | 0.73 |
| svmguide3 | 346.4±11.6 | **297.2±2.0** | 297.5±1.5 | 0.72 |
| magic04 | 6152.4±54.7 | 5536.7±59.3 | **5382.8±98.9** | 0.68 |
| a8a | 8993.8±40.3 | **6710.7±117.8** | 7313.5±277.7 | 0.75 |

for each auxiliary feature whereas Aux-Drop can handle that feature seamlessly using only one node. In numbers, if we assume 200 auxiliary features, 200 hidden nodes and 800 nodes in the AuxLayer then Aux-Net has about 32M more parameters than Aux-Drop.

**Aux-Drop settings**  Aux-Net is compared with Aux-Drop(ODL) and its settings are kept similar to Aux-Net for a fair comparison. Aux-Drop(ODL) is trained with 11 hidden layers, considering the $3^{rd}$ hidden layer as AuxLayer. Each layer has 50 nodes and the AuxLayer has 100 nodes. The smoothing rate is set as 0.2, the discount rate is fixed at 0.99 and the dropout is chosen as 0.3. We use cross-entropy loss. The learning rate is 0.3. Since the number of instances is less, a higher learning rate helps the model converge faster.

**Result**  We report the average loss by taking the mean of the total loss of each instance over the whole dataset. Aux-Drop(ODL) is run 20 times with random seeds for all the different values of $p$ and the average loss $\pm$ standard deviation is calculated. The result is shown in Table 3. Aux-Drop(ODL) outperforms Aux-Net in all seven different probability scenarios. In the situation, where $p = 0.9, 0.95, 0.99$, the difference in performance between Aux-Drop(ODL) and Aux-Net is very less. It is because of the less haphazardness in the data, the efficiency of the auxiliary dropout was not used to its fullest. Whereas, when the haphazard inputs are very high and frequent ($p = 0.8, 0.7, 0.6, 0.5$), the difference between Aux-Net and Aux-Drop(ODL) is high. Here, dropout makes the features independent of each other and hence when the features are not available frequently, it doesn't affect the performance of the model. Whereas, Aux-Net has dedicated layers for each auxiliary feature, requiring time to converge whenever the data is infrequent. The change in performance in the case of $p = 0.8, 0.7, 0.6$ and $0.5$ is 13.53%, 14.52%, 19.02% and 13.2% respectively. The amount of haphazardness is highest when $p = 0.5$, implying the effectiveness of Aux-Drop(ODL).

## 4.2 Comparison with state-of-the-art OLVF

We consider datasets with enough instances ($\geq 1000$) and variability from the OLVF paper to apply our model. We chose 4 different UCI datasets, namely, german, svmguide3, magic04 and a8a to simulate the scenarios of haphazard inputs. We compare our model with OLVF for the scenarios of 0.25 removing ratio ($Rem$) where it denotes that 25% of the instances are removed. We consider the first 2 features as base features and the remaining features as auxiliary features in all the datasets. For a fair comparison, we simulate the same amount of haphazardness as OLVF. For e.g., in the magic04 dataset with 10 features, $Rem = 0.25$ in OLVF experiments accounts for $Rem = 0.32$ for 8 auxiliary features in Aux-Drop. The probability $p$ of the availability of each auxiliary input is $1 - Rem$, therefore, $p = 0.68$ for the magic04 dataset. Similarly, we calculate the value of $p$ for each dataset and round it to two decimal places. The $p$ value in the Aux-Drop for each dataset is shown in Table 4. We compare OLVF with both Aux-Drop(ODL) and Aux-Drop(OGD).

**Aux-Drop Settings**  Both Aux-Drop(ODL) and Aux-Drop(OGD) have the same setting and hence are collectively referred to as Aux-Drop. Since the number of instances is less, we design Aux-Drop with only 6 hidden layers. The third layer is set as the AuxLayer. Each hidden layer has 50 nodes but the number of nodes in AuxLayer is different for each dataset considering the number of features and dropout value. The dropout value is set as 0.3. The number of nodes in AuxLayer is 100 except for a8a which has 400 nodes in AuxLayer. The number of auxiliary features in a8a is 121 and the dropout value is 0.3, so we need about

**Table 5:** Experiments on the trapezoidal data streams. Aux-Drop(ODL) and Aux-Drop(OGD) are compared with OLSF and OLVF (metrics reported from their original paper) in terms of the average number of errors. All the experiments are performed 20 times and the mean ± standard deviation is reported.

| Dataset | OLSF | OLVF | Aux-Drop(ODL) | Aux-Drop(OGD) |
|---------|------|------|---------------|---------------|
| german | 385.5±10.2 | 329.2±9.8 | **312.2±8.0** | 320.9±39.4 |
| svmguide3 | 361.7±29.7 | 351.6±25.9 | **296.9±1.0** | 297.0±0.9 |
| magic04 | 6147.4±65.3 | 5784.0±52.7 | 6361.25±319.6 | **5635.8±100.9** |
| a8a | 9420.4±549.9 | 8649.8±526.7 | 7850.9±15.9 | **7848.8±10.3** |

(121/0.3 ∼ 403) nodes. The smoothing rate for Aux-Drop(ODL) is 0.2 and the discount rate is 0.99. The cross-entropy loss is employed to train the model. The learning rate is 0.1 for the smaller datasets, i.e., german and svmguide3, whereas, for the larger datasets, i.e., magic04 and a8a, it is set as 0.01.

**Result** We consider the metric reported by OLVF and calculate the number of errors for each dataset. Aux-Drop is run 20 times randomly with respect to data shuffling, creating haphazard inputs and initializing the model as done in the OLVF manuscript. The mean and the standard deviation of these 20 experiments are reported in Table 4. Aux-Drop outperforms OLVF for all the datasets. Aux-Drop(ODL) performs better than Aux-Drop(OGD) in all datasets except magic04. The worst-performing experiment out of the 20 experiments of Aux-Drop is better than the best-performing experiment of OLVF.

### 4.3 Experiments on trapezoidal data streams

We experiment on the trapezoidal data streams and compare them with the OLVF and OLSF (best performing) algorithms Zhang et al. (2016). The trapezoidal streams are simulated by splitting the data into 10 chunks. The number of features in each successive chunk increases with the data stream. The first chunk has the first 10% of the total features, the second chunk has the first 20% features, and so on. The Aux-Drop setting and the dataset used are similar to the section 4.2 except the number of nodes in AuxLayer for a8a is 600. Only the first two features are considered the base features. Note that first ∼12 features (10%) are always available in this case. All the models are run 20 times and the mean ± standard deviation is reported.

**Result** Table 5 shows the performance of Aux-Drop as compared to others. The Aux-Drop outperforms OLVF and OLSF in all the datasets except magic04. The mean error is low for Aux-Drop compared to OLVF and OLSF. The amount of error in Aux-Drop(ODL) and Aux-Drop(OGD) compared to OLSF and OLVF is 16.7% and 9.2% less in the a8a dataset, suggesting that as the amount of data increases, the performance of Aux-Drop massively increases as compared to OLSF and OLVF. Furthermore, the standard deviation of Aux-Drop is low from OLVF and OLSF in all the datasets (except magic04), showing that the Aux-Drop has performed well consistently in all 20 experiments.

### 4.4 Evaluation on big datasets

HIGGS and SUSY dataset is used by ODL to report its metrics. Hence, we found them suitable to test the performance of Aux-Drop(ODL). We run our experiment for the first 1M instances. We design two experiments on this dataset: (1) Experiment on variable probability as done in section 4.1, and (2) Experiment on obsolete and sudden unknown features. Since the HIGGS and SUSY dataset is large, we run all the experiments 5 times (instead of 20) and the average is reported. For comparison, we chose ODL as the base model trained with only base features and refer it as ODL(B). For HIGGS and SUSY, we consider the first 5 and 2 features as base features and the next 16 and 6 features as auxiliary features, respectively.

**AuxDrop Settings** In both the cases of HIGGS and SUSY, we train the Aux-Drop(ODL) with 11 hidden layers. The 3rd layer is set as the AuxLayer. The number of neurons in each hidden layer is 50 and in the AuxLayer is 100. The dropout value is 0.3 and the learning rate value is set at 0.05. The discount rate is fixed at 0.99 and the smoothing rate is 0.2. For a fair comparison, we design ODL with 11 hidden layers, 50 nodes in each hidden layer and the same value of learning rate, discount rate and smoothing rate.

**Table 6:** Error in HIGGS and SUSY for various probability $p$. The metric is reported as the mean ± standard deviation of the number of errors in 5 runs. The error of ODL(B) trained on only base features for HIGGS and SUSY dataset is 441483.2±184.3 and 286198.6±189.4, respectively. The ΔAvg value reported in the table is calculated by subtracting the average number of errors of ODL(B) with the average number of errors of Aux-Drop(ODL), respectively.

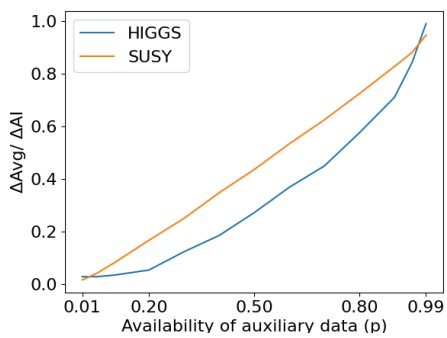

| $p$ | HIGGS | | SUSY | |
|---|---|---|---|---|
| | Avg±Std | Δ Avg | Avg±std | Δ Avg |
| .01 | 440033.4±129.9 | 1449.8 | 285088.0±69.3 | 1110.6 |
| .05 | 440045.4±250.5 | 1437.8 | 283463.0±305.5 | 2735.6 |
| .10 | 439752.0±198.5 | 1731.2 | 280752.4±396.2 | 5446.2 |
| .20 | 438775.2±361.7 | 2708.0 | 274907.0±575.7 | 11291.6 |
| .30 | 435286.0±675.5 | 6197.2 | 269269.8±549.6 | 16928.8 |
| .40 | 432190.8±381.3 | 9292.4 | 262713.4±632.3 | 23485.2 |
| .50 | 427844.8±616.1 | 13638.4 | 256719.4±618.3 | 29479.2 |
| .60 | 423002.8±604.5 | 18480.4 | 250108.0±829.5 | 36090.6 |
| .70 | 418927.4±495.2 | 22555.8 | 243954.2±813.7 | 42244.4 |
| .80 | 412601.6±254.0 | 28881.6 | 237211.6±654.5 | 48987.0 |
| .90 | 405834.6±350.3 | 35648.6 | 230216.2±698.5 | 55982.4 |
| .95 | 399234.8±613.7 | 42248.4 | 226631.8±354.4 | 59566.8 |
| .99 | 391787.8±641.8 | 49695.4 | 222151.6±181.4 | 64047.0 |

**Figure 4:** The average error and standard deviation of ODL on the whole features (base + auxiliary features) for 5 runs for HIGGS and SUSY are 391334.8 and 218622.2. Thus the performance improvement (ΔAI) of ODL(B) because of the addition of the whole auxiliary features are 50148.4 and 67576.4 for HIGGS and SUSY, respectively. The fraction improvement in the Aux-Drop(ODL) is calculated by the ratio of ΔAvg (in Table 6) and ΔAI. The value of $p$ (in Table 6) denotes the amount of auxiliary information available.

### 4.4.1 Experiment on variable probability

We vary the availability of each auxiliary feature by a uniform distribution of probability $p$. Each auxiliary feature is varied by the same value of $p$, but they are independent of each other. We consider all the situations such as when very little auxiliary data is present ($p = 0.01$), the haphazardness in the data is maximum ($p = 0.5$), almost all the auxiliary data is available all the time ($p = 0.99$), etc.

**Results** The mean and the standard deviation on HIGSS and SUSY are shown in Table 6. The mean and standard deviation of the number of errors of ODL(B) for HIGGS is 441483.2 and 184.3 respectively. Whereas, for the SUSY, it is 286198.6 and 189.4, respectively. We consider this value as the base value and also report the metrics for all the $p$ values with respect to this in Table 6. The Δ Avg reported in the table shows the less amount of errors made by Aux-Drop(ODL) utilizing the extra information from auxiliary features and is calculated by subtracting the average number of errors of ODL(B) from the average number of errors of Aux-Drop(ODL), respectively. Aux-Drop(ODL) is able to incorporate even a little amount of data from auxiliary features when $p = 0.01$ and gives better performance. Moreover, at each increasing $p$ value, the Aux-Drop(ODL) performance improves. This is better represented in Figure 4 where the progression of the fraction improvement (ΔAvg/ΔAI) is shown with respect to the availability of auxiliary data ($p$). The average error of ODL on all available datasets (i.e., all the auxiliary features are always available) is also reported here. For HIGGS, ODL trained on the 21 features gives an error of 391334.8 whereas, for SUSY, it is 218622.2. Based on this, we can say that the performance improvement (Δ AI) achieved by ODL(B) due to auxiliary data for HIGGS and SUSY is 50148.4 and 67576.4, respectively.

### 4.4.2 Obsolete and Sudden Unknown features

We demonstrate the effectiveness of Aux-Drop(ODL) in processing the extra information received from auxiliary features in both the SUSY and HIGGS datasets. Here, we design the data in a such way that all of them are sudden features, i.e., there is no information about the existence of these features when the model is defined. The model knows about this feature suddenly at time $t$ after the model deployment. For the SUSY dataset, the first auxiliary feature starts arriving from 100k till 500k, the next auxiliary feature ranges from 200k till 600k, and so on to the 6th auxiliary feature coming from 600k to 1000k instances.

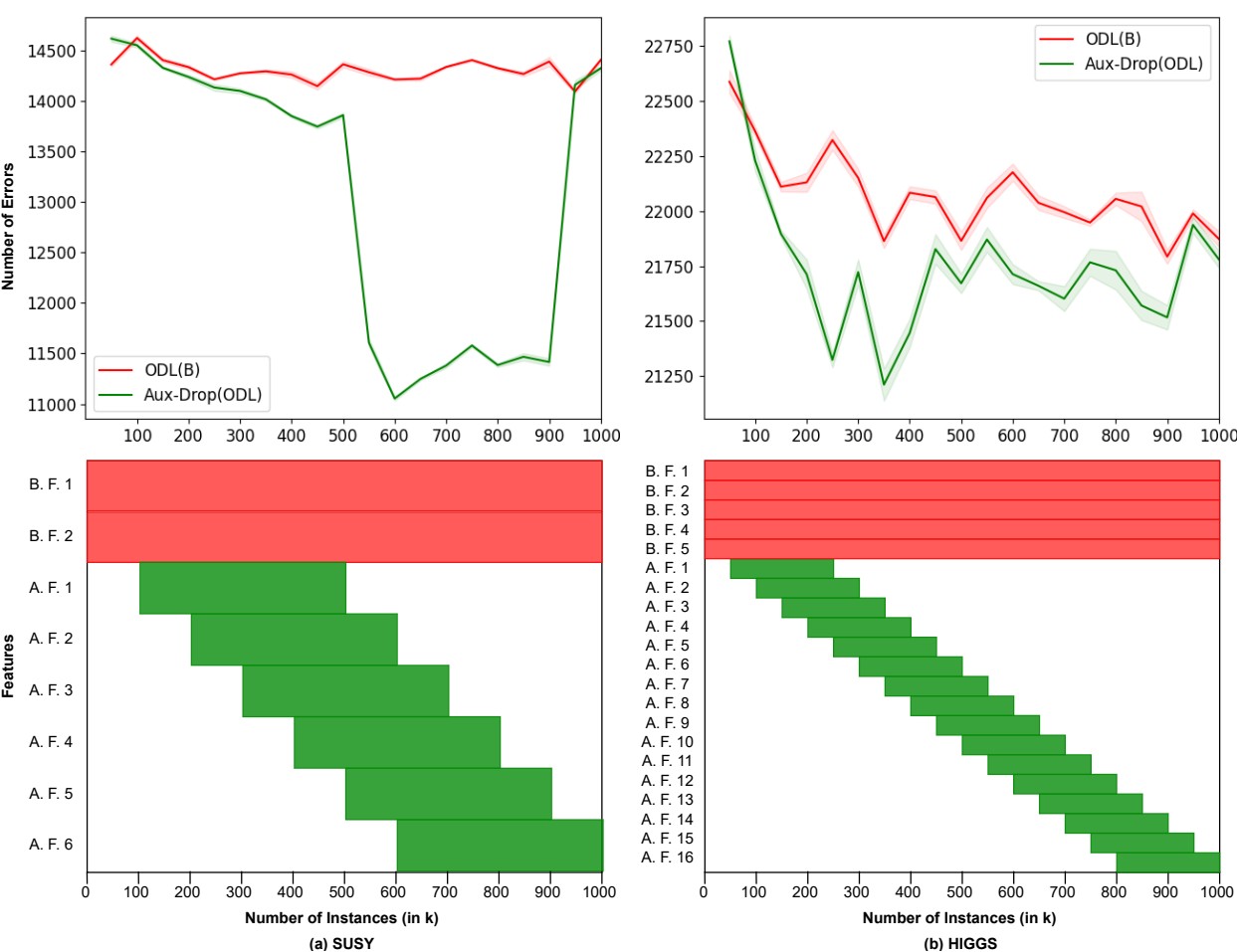

**Figure 5:** Result of the obsolete and sudden features experiment on (a) SUSY and (b) HIGGS dataset for Aux-Drop(ODL). Every experiment is run 5 times and the average value is reported. We calculate the average number of errors in each 50k instance. Thus, the graph starts at 50k and goes till 1000k. B. F. {n} and A. F. {n} represent the $n^{\text{th}}$ number base features and auxiliary features, respectively. 95% confidence interval (CI) is also shown in the graph. The green box represents the time instances when a certain auxiliary feature was available. The x-axis denotes the number of instances (in k).

Each feature becomes obsolete after arriving for 400k instances. Similarly for the HIGGS dataset, the first auxiliary feature arrives from 50k to 250k instances, the second arrives from 100k to 300k, and so on where every successive auxiliary feature arrives at 50k instances after the previous auxiliary features start arriving and arrive till the next 200k instances. This is better depicted in the lower part of Figure 5(a) for SUSY and Figure 5(b) for HIGGS.

**Results** Figure 5 shows the result of obsolete and sudden unknown features for both SUSY and HIGGS. The performance of ODL(B) and Aux-Drop(ODL) is similar for the first 50k instances since both get the same amount of data. But as Aux-Drop(ODL) gets the auxiliary information, its performance improves. The maximum amount of auxiliary information received in the SUSY dataset is from 400k to 700k and we see that the best performance is during that period. The minimum is achieved at 600k. Moreover, in the later stages after 900k, when the auxiliary information reduces, the Aux-Drop(ODL) converges to the performance of ODL(B) depicting the agile manner in which Aux-Drop(ODL) handles the haphazard inputs. In the case of HIGGS, Aux-Drop(ODL) is better than ODL as soon as it starts getting the auxiliary information. For each period of 50k instances, we get a confidence interval (CI) low and CI high. We calculate the 95% CI. The average of all these 20 periods gives a CI low of 13027.9 and a CI high of 13077.2 for Aux-Drop(ODL) and a

**Table 7:** Table shows the need for AuxLayer and the use of dropout. Notations: RDANDO - Random Dropout in AuxLayer No Dropout in Others, RDAL - Random Dropout in All Layers, ADARDO - Auxiliary Dropout in AuxLayer and Random Dropout in Other layers, RDIFL - Random Dropout in First Layer with all features passed directly to the first layer. The probability $p = 0.27$, 0.28, 0.32 and 0.25 for german, svmguide3, magic04 and a8a datasets respectively.

| Methods | german | svmguide3 | magic04 | a8a |
|---|---|---|---|---|
| RDANDO | 318.0±2.8 | 297.6±1.9 | 6123.5±169.1 | 7853.1±16.3 |
| RDAL | 319.3±4.4 | 298.2±3.0 | 6433.1±143.7 | 7862±323.2 |
| ADARDO | 318.6±4.2 | 297.6±1.7 | 6700.4±33.1 | **7852.9±15.9** |
| Aux-Drop(ODL) | **317.4±1.9** | **296.9±1.5** | **6039.1±190.4** | 7855.5±16.8 |
| RDIFL | 318.5±2.8 | 297.2±1.7 | 6528±136.4 | 7869.9±33.2 |

CI low of 14289.1 and a CI high of 14332.0 for ODL in the SUSY dataset. Whereas for the HIGGS dataset, the average CI (low, high) is (21696.8, 21799.4) for Aux-Drop(ODL) and (22039.1, 22109.2) for ODL.

## 5 Ablation Studies

For all the ablation studies, we use Aux-Drop(ODL) framework and apply it to the four UCI datasets (german, svmguide3, magic04 and a8a). We prepare all the haphazard datasets following the subsection 4.2. In order to increase the complexity, we perform all the experiments with less availability of auxiliary features. Hence we consider 1-$p$ value from Table 4. Thus, the probability of the availability of auxiliary features is 0.27, 0.28, 0.32 and 0.25 for german, svmguide3, magic04 and a8a datasets respectively. The architecture of Aux-Drop(ODL) is also followed from subsection 4.2 until it is specified otherwise. Aux-Drop(ODL) has 6 hidden layers with the 3rd layer as the AuxLayer. In this section, the terms Aux-Drop(ODL) and Aux-Drop are used interchangeably.

### 5.1 Need of AuxLayer

One of the requirements of Aux-Drop is the presence of alteast one base feature. So, we design a model where we pass all the inputs (base and auxiliary features) directly to the first layer itself without the use of AuxLayer in ODL. Here, we employ Random Dropout in the First layer to handle the haphazard inputs (RDIFL). The performance of RDIFL is shown in the lower part of Table 7. It can be seen that Aux-Drop(ODL) outperforms RDIFL by 7.4% in magic004 and is marginally better in other datasets. This is because Aux-Drop utilizes the full information from base layers and increments it with the information from haphazard auxiliary features.

### 5.2 Effective Use of Dropout

We apply the dropout in the AuxLayer with an emphasis on the coupling between the auxiliary feature and auxiliary node by the manner of selectively choosing nodes to drop, based on the unavailability of auxiliary features. But, it is to be noted that, dropout can be applied randomly in the AuxLayer too. Moreover, dropout can also be applied to the other hidden layers as well. We present a comparison of all these ways of applying dropout and show empirically that Aux-Drop is the best way to employ dropout. We compare Aux-Drop with its three other variants: (a) RDANDO - Random Dropout is applied in the AuxLayer and No Dropout is applied in Other layers, (b) RDAL - Random Dropout is applied in All Layers, and (c) ADARDO - Auxiliary Dropout is applied in the AuxLayer and Random Dropout is applied in all the Other layers. The results of all these methods (applied on ODL) are compared with Aux-Drop (ODL) in the 4 UCI machine learning dataset and are shown in the upper half of Table 7. Aux-Drop is better in all the cases except in a8a. The maximum variation in the results is seen in the magic04 dataset. The second best method is RDANDO which also applies dropout only in the AuxLayer. So, the best way is to employ auxiliary dropout only in the AuxLayer.

**Table 8:** Comparison of the position of AuxLayer. Pos here stands for the position. The experiment is conducted on the four UCI datasets. The probability $p = 0.27$, $0.28$, $0.32$ and $0.25$ for german, svmguide3, magic04 and a8a datasets respectively.

| Pos | german | svmguide3 | magic04 | a8a |
|-----|--------|-----------|---------|-----|
| 1 | 317.7±2.0 | 298.5±2.9 | 6049.4±269.5 | 7842.1±31.6 |
| 2 | 319.1±3.5 | 298.4±2.8 | 6054.0±213.3 | **7730.5±73.5** |
| 3 | **317.4±1.9** | **296.9±1.5** | **6039.1±190.4** | 7855.4±16.8 |
| 4 | 318.5±2.7 | 298.9±2.8 | 6110.7±146.2 | 7852.9±12.4 |
| 5 | 317.7±2.2 | 297.4±1.7 | 6428.7±105.5 | 7856.7±14.3 |

### 5.3 Effect of AuxLayer position in the Model

The position of AuxLayer is a hyperparameter in the Aux-Drop. In all the above methods, we fixed the $3^{rd}$ layer as the AuxLayer. Here, we demonstrate how the model performs with respect to the different positions of the AuxLayer in the 4 UCI datasets. The results are shown in Table 8. The $3^{rd}$ layer seems to be the best position except for a8a which gives the best performance for the $2^{nd}$ position. The ratio of base and auxiliary features is 1:60.5 for a8a. Thus, it requires comparatively more layers to process the auxiliary information. The $2^{nd}$ position outperforms the $1^{st}$ position in a8a because the first hidden layer helps to process the base features. Whereas for the other three datasets, the maximum ratio of base features and auxiliary features is 1:11 (for german) and hence comparatively less number of layers are enough to capture the auxiliary features.

## 6 Conclusion

The challenge and application of haphazard inputs are immense and to our knowledge, there are no effective deep learning methods available to handle it. So, we propose a generalized concept called Aux-Drop which can be applied to any deep learning-based online architecture. We demonstrate the effectiveness of Aux-Drop in multiple datasets and empirically assert the importance of the Aux-Drop design by applying it to ODL and OGD frameworks. The various experiments on big datasets meticulously show the agile manner in which Aux-Drop processes the auxiliary information and converges to the base deep learning architecture during the unavailability of auxiliary features.

**Acknowledgement** We acknowledge the various funding that supported this project: Researcher Project for Scientific Renewal grant no. 325741 (Dilip K. Prasad) and UiT's thematic funding project VirtualStain with Cristin Project ID 2061348 (Alexander Horsch and Dilip K. Prasad).

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

## A  Adversarial Inputs

Adversarial inputs are malicious inputs designed to cause the machine-learning model to make a mistake. This is achieved by modifying the input data subtly (Goodfellow et al., 2018). Machine learning models are highly vulnerable to small changes in the input at test time (Biggio et al., 2013; Kurakin et al., 2016). Hence dealing with adversarial inputs require special methods and training procedure like injecting adversarial examples into the training set (Huang et al., 2015), using defensive distillation for network training (Papernot et al., 2016). Whereas, haphazard inputs are not modified, disturbed, or perturbed. All the inputs are used as it is and no new inputs are created at any point in time.

## B   Semi-supervised techniques

Adaptations of generative adversarial networks (GAN) (Goodfellow et al., 2020) have achieved competitive results in semi-supervised learning (Liu & Xiang, 2020). Approaches like MARK (Patil et al., 2022) attempt to use GAN to generate data that can exhibit the patterns of the unknown test data and this generated data in turn can train the model to deal with adversarial inputs (Goodfellow et al., 2018) and unknown attacks. However, this does not apply to haphazard inputs, since it cannot deal with changing input dimensions and unavailable auxiliary features.

## C   Progressive Networks

Progressive networks (Rusu et al., 2016) are designed to handle a complex sequence of tasks (Task$_1$, Task$_2$, ..., Task$_k$) by leveraging transfer learning and avoiding catastrophic forgetting. Here, at Task$_1$ a deep neural network ($D_1$) is trained in an offline manner. When Task$_2$ arrives, a copy of $D_1$ is made with random initialization (let's say $D_2$), and $D_1$ is frozen. The input to $D_2$ is now the hidden activations from $D_1$ and the actual input. This goes on till Task$_k$. Whereas in the haphazard inputs problem, there is only one task without any change and the model is trained in an online manner with a stochastic gradient. Hence, progressive networks can not be applied here. One can still think of a connection between progressive networks and Aux-Drop by considering each new auxiliary feature as task-specific and creating a deep neural network whenever a new feature arrives. In a practical scenario, the number of auxiliary features can go up to thousands (121 in the case of the a8a dataset) which will require 1000 deep neural networks. This is not feasible from both the point of storage and computation.

## D   Domain Adaptation

Domain adaptation aims to train a model using source data (training data) in such a way that the model generalizes well to the target data (test data) too (Farahani et al., 2021). Training and test data can be from different distributions, hence it becomes an important part of the model to explicitly handle the data from different domains (distribution). Domain adaptation is a special case of transfer learning (Pan & Yang, 2010). Interestingly, domain adaptation has only been applied in the online learning setting (Chen et al., 2021). But the biggest assumption in domain adaptation is that all the features are base features and hence it cannot handle the haphazard input problems.

## E   Open Set

Open set problems deal with scenarios where the number of classes (output labels) is unknown at the training time and new classes can appear during testing (Scheirer et al., 2012). The haphazard inputs problem is orthogonal to the open set problem since the number of classes is always known and the number of input features is unknown.

