# OpenReview forum: "Aux-Drop: Handling Haphazard Inputs in Online Learning Using Auxiliary Dropouts"
_TMLR — Accepted by TMLR_

### Review · Reviewer_7CMD · 2023-03-30

**Summary Of Contributions:**

The paper addresses the problem of online learning with features that can have time-variable size and distribution. New unknown features may appear, old features may disappear and become obsolete. The problem is handled via a dropout-inspired  AUX layer proposed in the paper.

**Audience:**

Yes

**Claims And Evidence:**

No

**Requested Changes:**

- I formulated some core change requests in the previous section, because they are tightly linked with weaknesses, please do not overlook
- There is a clear link between the proposed method and the semi-supervised techniques based on filling the blanks, including the Pseudolikelihood (see Section 18.3 in the Deep Learning book by Goodfellow et al.) methods for example. It would be nice to have this link reflected in the literature review and I would encourage authors to also formulate a probabilistic model of their approach to strengthen the theoretical foundation of their paper.
- Please discuss in detail how your method relates to Progressive networks, domain adaptation approaches and open set problems
- Please provide reference for Italy Power Demand dataset
- In Table 3 caption, please correct typo "Aux-Drop is run 5 fives times"
- In Table 3, can you explain why the metric is changing abruptly with increasing $p$? What is the statistical significance of results? How many runs did you use to average the results? If it's only one run, the results presented are not statistically significant. Please run multiple seeds and present average results with confidence intervals. I mean this is logical, especially that you do follow a more rigorous statistical protocol in Tables 4,5.
- ODL is an integral part of the proposed approach. However, the details of the method are not covered in the paper. Even though this is not original contribution of this paper, I believe it would be beneficial for readability reasons to provide a section that explains the ODL and how exactly it is used in conjunction with the proposed approach.
- In Figure 4, please provide curves averaged over several runs along with confidence intervals (semi-transparent, same colour as the curve)
- According to Table 8, the position of Aux layer has very little effect on performance. How can you explain it? What happens when you put it in position 1? Also in previous experiments you use 11 layers, and in this one only 6. Why? What happens if you use 11 layers?
- I believe Appendices A and B provide important information and should be moved to the main body of the paper.

**Strengths And Weaknesses:**

STRENGTHS
- Very interesting paper and a relevant topic
- The motivation of technical contribution clear, I especially like Table 1
- Empirical results are based on several datasets, adding credibility
- The work seems to be sufficiently novel and contribution is non-trivial


WEAKNESSES
- The general motivation is rather weak. There is a lot of emphasis on the self-driving cars, aerospace and real applications. However, the datasets explored in the experimental section do not cover these areas at all and they are simulated. I suggest that the authors rewrite the intro significantly, focusing it specifically on the kinds of problems that they really solve and also make sure that the claims they make are actually supported by the datasets they use in experiments.
- Four contributions are listed in the paper, but I find that only first two are the actual science contributions, whereas the remaining two points, which I propose to remove, simply describe the properties of the proposed method.
- Additionally, the second contribution claims that the proposed technique can be seamlessly combined with existing neural networks. However, the experiments only cover the coupling of one neural architecture with the proposed layer. I request that the author provide experiments supporting this claim by demonstrating that it works with at least one more architecture that is differnt from the one they use in the current experimental set.
- The description of the datasets is very poor in the experimental section. I had to go to the original dataset pages to learn that they are Monte-Carlo simulated and those pages do not contain enough information about the nature of the problem simulated. Please add a motivation subsection in the experiments section clearly describing the datasets and motivating why the use of these datasets is acceptable to support the claims put forward in the paper. The statement that a particular dataset was used in a previous paper is not a good motivation - please take this into account - I will not accept this excuse.
- Related work section presents a number of relevant methods, but the experimental results compare against only two of them. Why is this the case? Could you please provide more comparisons to make results more convincing?
- The method is clearly related to group dropout, Ke et al., Group-Wise Dynamic Dropout Based on Latent Semantic Variations, AAAI 2020, reducing the originality of the method. Could you please explain the novelty of your method with respect to dropout techniques utilizing grouping?
- The paper provides no code, reproducibility is compromised. Could you please release the code?

Overall, I'm optimistic about the paper outlook, however this is subject to authors delivering a thorough revision of the paper based on all the points I raised. Therefore, I do encourage the authors to provide a comprehensive revision of the paper.

---

> ### Author Response · Authors · 2023-04-20
>
> We thank the reviewer for their thorough comments and try to resolve them as much as possible. Due to the number of characters limitation we specify your comments with the beginning of your written sentence and ending it with "...". and we address the comments in two separate comments. This comment handles the weakness mentioned by you and the next comment handles the requested changes mentioned by you.
>
> 1. The general motivation is rather weak ...
> - Thank you for your constructive comments. Yes, we agree with you that there is a lack of datasets for haphazard input as well as the formal problem statement is also lacking. So we have tried to formalize the problem statement in the revised manuscript. We have excluded possible future applications discussion of this manuscript problem statement now. Hence, we have rewritten the introduction so that these applications are not discussed (see 1st paragraph of introduction, Page 1). We believe more research activity in this novel problem statement will help us move forward in this domain.
>
> 2. Four contributions are listed in the paper, but ...
>  - Thank you for your comments. As suggested, we have concise the contributions (see the last paragraph in section 1, page 3).
>
> 3. Additionally, the second contribution claims ...
>  - We have now included online gradient descent (OGD) architecture coupled with Aux-Drop in the paper and refer to it as Aux-Drop(OGD). OGD is a deep-learning model capable of handling streaming data [1]. The motivation to use OGD comes from the ODL [1] paper, where they compared their model with OGD. In our paper, see the "Invariant to Architecture" paragraph in section 3.4 (Page 8) for the architectural difference between OGD and Aux-Drop(OGD) and sections 4.2 (Page 12) and 4.3 (Page 13) for the performance of Aux-Drop(OGD).
>
>  4. The description of the datasets is very poor ...
>  - The description of all the datasets can be found in the Datasets paragraph of section 4 (see blue colored text in the dataset paragraph, page 9 and 10). The motivation to choose these datasets is two-fold: (1) The first is to see how Aux-Drop performs on small (german, Italy Power Demand, svmguide3), medium (magic04, a8a) and large (SUSY, HIGGS) datasets. (2) The number of features in the dataset ranges from 8 (SUSY) to 123 (a8a). This helps in demonstrating the flexibility of Aux-Drop in the varying number of auxiliary features. This is also included in the "Motivation of the Chosen Datasets" paragraph in section 4 (page 10).
>
>  5. Related work section presents a number ...
>  -  We compare our model with other related works with which our problem statement matches fully or partially. Aux-Net, OLVF, OLSF and ODL are used for comparison. Aux-Net is used to give a comparison with a deep-learning approach. OLVF can handle haphazard inputs and hence is chosen as a baseline. OLSF can handle a trapezoidal data stream which is a subset of the haphazard input problem and hence is included in our study. We have contacted the authors of OCDS [2] for implementation details about their work or the availability of OCDS code but no response was received. Due to a lack of code availability, we are unable to compare with OCDS.
>
> 6. The method is clearly related to group dropout, ...
> - We now include a paragraph "Dropout" in the related works section (see section 2, Pages 3 and 4) which discusses prominent dropout-based methods available in the literature. Group-wise dropout [3] adjusts the dropout probabilities adaptively using the feature density by analyzing the number of linearly uncorrelated deep features gathered over equally spaced grids in low-dimension feature space. All the discussed dropout strategy in the literature including Group-wise dropout is employed to better generalize the model and prevent overfitting but none of these strategies can be used to handle haphazard inputs. Aux-Drop drops the nodes in the hidden layer based on the unavailability of auxiliary features and dropout rate.
>
> 7. The paper provides no code ...
> - The code of Aux-Drop can be found in the supplementary file.
>
> References
>
> [1] Doyen Sahoo, Quang Pham, Jing Lu, and Steven CH Hoi. Online deep learning: Learning deep neural networks on the fly. arXiv preprint arXiv:1711.03705, 2017.
>
> [2] Yi He, Baijun Wu, Di Wu, Ege Beyazit, Sheng Chen, and Xindong Wu. Online learning from capricious data streams: a generative approach. In International Joint Conference on Artificial Intelligence Main track, 2019.
>
> [3] Zhiwei Ke, Zhiwei Wen, Weicheng Xie, Yi Wang, and Linlin Shen. Group-wise dynamic dropout based on latent semantic variations. In Proceedings of the AAAI Conference on Artificial Intelligence, volume 34, pp. 11229–11236, 2020.

---

> ### Author Response · Authors · 2023-04-20
>
> This comment handles the requested changes mentioned by you. See another comment for the weakness mentioned by you.
>
> 8. There is a clear link between the ...
> - Semi-supervised approaches like MARK [4] attempt to use GAN to generate data that can exhibit the patterns of the unknown test data and this generated data in turn can train the model to deal with adversarial inputs and unknown attacks. However, this does not apply to haphazard inputs, since it cannot deal with changing input dimensions and unavailable auxiliary features. We include the above differences in the "Haphazard Inputs and Aux-Drop vs Other Works" paragraph (see last paragraph in section 3.4, Page 9) and Appendix B (Page 20).
> - Pseudolikelihood - Our problem statement is a closer match with the Bernoulli distribution because it is a discrete probability distribution that models the probability of a binary outcome for a single trial. Whereas pseudolikelihood requires the assumption of conditional independence between the variables but for our problem statement conditional independence may not hold because the presence or absence of one feature may affect the probability of another feature being present.
> - Thank you for your feedback about including the theoretical foundation in the paper. We have included a theoretical foundation based on Bernoulli's distribution (see Auxiliary dropout paragraph in section 3.2, page number 6).
>
> 9. Please discuss in detail ...
> - Please see "Haphazard Inputs and Aux-Drop vs Other Works" paragraph (see last paragraph in section 3.4, page 9) and Appendix C (Page 20) for progressive networks, Appendix D (Page 21) for domain adaptation approaches and Appendix E (Page 21) for open set problems.
>
> 10. Please provide reference for Italy Power Demand dataset
> - See the blue-colored citation in the first line of section 4 (Page 9) and the 1st link in the footnote of page 9.
>
> 11. In Table 3 caption, please correct typo "Aux-Drop is run 5 fives times"
> - The typo is corrected and moreover, the results are now reported for the average of 20 runs (see blue-colored text in the caption of Table 3, page 11) to keep it consistent with other experiments in sections 4.2 and section 4.3.
>
> 12. In Table 3, can you explain ...
> - We re-run all the experiments in Table 3 to make them consistent with Table 4 and Table 5. We performed 20 runs of the model and report the standard deviation of each experiment as done in Table 4 and Table 5. As the value of p increases, the amount of data received by the model increases, thus the performance of the model improves. Please see Table 3 (blue-colored values) on Page 11 for the changes.
>
> 13. ODL is an integral part ...
> - The brief description of ODL along with its coupling with Aux-Drop is presented in the "Invariant to the Architecture" paragraph in section 3.4 (page 8). A visual change in the architecture of ODL when coupled with Aux-Drop is also presented in Figure 3 (see page 9).
>
> 14. In Figure 4, please provide ...
> - We believe there is a typo in this comment. It should have been Figure 3 instead of Figure 4 in the old version of the manuscript. Anyways, we are considering this comment regarding Figure 3. The numbering of Figure 3 in the old manuscript has now become Figure 5 in the new manuscript. We have included the 95\% confidence interval in Figure 5 (see page 14) . We also report the average confidence interval values in the text (see blue-colored text in section 4.4.2, page 15).
>
> 15. According to Table 8, the position ...
> - The position of AuxLayer has little effect on small datasets (german and svmguide3) but for medium size datasets (magic04 and a8a) the choice of initial layer gives better performance. It is because more layers are available to the model to process the maximum amount of data (i.e., auxiliary features). The result with 1st layer as AuxLayer is also included in Table 8 (page 16). It gives comparative results but does not outperform Aux-Drop with the 3rd layer as AuxLayer. We can conclude that the initial hidden layers are a better choice for AuxLayer but within those initial layers (1, 2, 3), there is not much difference. It is to be noted that Aux-Drop is always run with 6 hidden layers for small datasets (german and svmguide3) and medium-size datasets (magic04 and a8a) in all the experiments in the paper. To avoid confusion, we now include a paragraph in section 5 (first paragraph, pages 15 and 16) stating the same.
>
> 16. I believe Appendices A and B ...
> - Thank you for pointing this out. Both of them have been moved to the main body of the paper. Appendix A is moved to the 1st paragraph of section 1 (page 1) and Appendix B is moved to section 3.3 (see pages 6, 7 and 8).
>
> References
>
> [4] Rajendra Patil, Vinay Sachidananda, Hongyi Peng, Akshay Sachdeva, and Mohan Gurusamy. Mark: Fill in the blanks through a jointgan based data augmentation for network anomaly detection. Computers \& Security, 119:102759, 2022.

---

> ### Comment · Reviewer_7CMD · 2023-04-24
> **Post rebuttal**
>
> I thank the authors for their effort in revising the manuscript. I am very satisfied with the revisions they implemented. The only remaining problem in my view is the code that they provided. The analysis is based on this repo: https://github.com/Rohit102497/Aux-Drop. All table and figure references are made to the old version of the manuscript.
>
> A number of issues have been identified with the code it seems important that the issues are fixed to ensure that (i) the paper's results are reproducible and (ii) other researchers can build on this work thereby increasing the impact of the paper and of the TMLR journal. I present the detailed analysis of code issues below and I hope that the authors can fix them before the final decision is made.
>
> - the code base supports 4 out of 7 datasets from Table 2 of the preprint: german, svmguide3, magic04, a8a. These all have 32500 instances or fewer, the big datasets such as SUSY and HIGGS with 1 million instances are missing.
> Italy Power Demand dataset and experiment code are not in the code base.
>
> - All experiments and figures on HIGGS and SUSY cannot be reproduced. We neither have the experimental code nor the data loaders. This includes Table 6 and Fig. 2 & 3.
>
> - Some model baseline code is missing (Aux-Net for Table 3 and ODL(B) for Table 6). So, tables 3 and 6 cannot be reproduced.
>
> - I was able to reproduce Table 4. Table 5 is similar to Table 4, but the experiment requires a 'trapezoidal' data loader. This is not provided, we only have data loaders for the experiment as formulated for Table 4 so it can be unclear what the exact inputs to the algorithm are.
>
> Based on this, I have the following requests:
>
> - Could you please provide data loaders for all datasets mentioned in the paper? (Italy power dataset, SUSY and HIGGS)
>
> - It seems that when I run AuxDrop_OGD the code fails to run. Could you please fix it?
>
> - Could you please provide experiment code to reproduce tables 3,5,6? This includes model, training and evaluation pipelines.

---

### Review · Reviewer_MXSX · 2023-03-31

**Summary Of Contributions:**

The paper proposes Aux-Drop that aims to address challenges in online learning coined as “haphazard inputs.” The proposed approach is based on dropout and is applied to the concatenated representations across base and auxiliary features. Missing auxiliary features are first dropped before dropout is applied to the remaining features. The proposed approach is evaluated on various datasets and demonstrate to outperform baselines such as OLSF and OLVF.

**Audience:**

No

**Claims And Evidence:**

No

**Requested Changes:**

1. The primary objective of this paper is ambiguous, as it is not clear whether the aim is to introduce a new problem setup or a novel learning algorithm. If the primary emphasis of the paper is on a new problem setup, it is crucial to highlight the differences between the proposed setup and existing ones, and provide a clear rationale for why the new setup is desired. On the other hand, if the main focus is on proposed a new algorithm, it is more preferable to begin the introduction by discussing the limitations of existing approaches.

2. The definition of "haphazard" lacks precision. It appears the notion of "haphazard" is intended to cover any potential problem that can arise in online learning, which makes the problem definition overly generic. The authors should clarify what is not considered as "haphazard" to better define the scope of the problem. If the proposed approach is intended to cover any online learning problem, does it include adversarial inputs?

3. Assumptions in empirical evaluation violates the broad scope of the problem definition. The definition of "haphazard" includes missing/obsolete/sudden-loss of features. But in the proposed approach and evaluation setup, the base features are always predefined, which contradicts with the flexible problem setup—does the proposed setup include assumptions in prior on which feature is always available?

4. Overall, I suggest the authors better define the problem/scope of this paper and try to prevent over generalizing the proposed approach.

**Strengths And Weaknesses:**

Strength:
1. The online learning setup that incorporates various challenges from the real world is interesting.
2. Competitive performance has been demonstrated by evaluating the proposed approach on multiple datasets of different sizes.

Weaknesses (detailed in the section below):
1. The primary objective of this paper is ambiguous, as it is not clear whether the aim is to introduce a new problem setup or a novel learning algorithm.
2. The definition of "haphazard" lacks precision. It appears the notion of "haphazard" is intended to cover any potential problem that can arise in online learning, which makes the problem definition overly generic. The authors should clarify what is not considered as "haphazard" to better define the scope of the problem.
3. Assumptions in empirical evaluation violates the broad scope of the problem definition.

---

> ### Author Response · Authors · 2023-04-20
>
> We thank the reviewer for their comments and try to resolve them as much as possible.
>
> 1. The primary objective of this paper is ambiguous, as it is not clear whether the aim is to introduce a new problem setup or a novel learning algorithm. If the primary emphasis of the paper is on a new problem setup, it is crucial to highlight the differences between the proposed setup and existing ones, and provide a clear rationale for why the new setup is desired. On the other hand, if the main focus is on proposed a new algorithm, it is more preferable to begin the introduction by discussing the limitations of existing approaches.
> - We accept that our writing has some limitations and thus we have rewritten the introduction part and we hope that the objective of the paper is clear now. The aim and the contribution of the paper are mentioned in the last paragraph of the introduction section (page 2).  We aim to formally introduce the problem of haphazard inputs since it has a very loose definition in the current literature. We also propose a novel concept based on dropout which can be applied to existing online learning-based deep-learning approaches to model haphazard inputs.
>
> 2. The definition of "haphazard" lacks precision. It appears the notion of "haphazard" is intended to cover any potential problem that can arise in online learning, which makes the problem definition overly generic. The authors should clarify what is not considered as "haphazard" to better define the scope of the problem. If the proposed approach is intended to cover any online learning problem, does it include adversarial inputs?
> - We now include a paragraph "Haphazard Inputs and Aux-Drop vs Other Works" in the discussion subsection (see section 3.4, page number 9) which discusses the difference between haphazard inputs with adversarial inputs among others. Adversarial inputs are malicious inputs designed to cause the machine-learning model to make a mistake. This is achieved by modifying the input data subtly. Whereas, haphazard inputs are not modified, disturbed, or perturbed. All the inputs are used as it is and no new inputs are created at any point in time.
>
> 3. Assumptions in empirical evaluation violates the broad scope of the problem definition. The definition of "haphazard" includes missing/obsolete/sudden-loss of features. But in the proposed approach and evaluation setup, the base features are always predefined, which contradicts with the flexible problem setup—does the proposed setup include assumptions in prior on which feature is always available?
> - Yes, we assume that there is atleast one feature always available (called base feature). We mention this now in the 4th paragraph (see text in blue, page 2) of the introduction section. We make sure that there is a fair comparison between the Aux-Drop and the baseline method. To achieve this, when 0.25 fraction of data is dropped from OLVF, 0.27, 0.28, 0.32 and 0.25 fraction of auxiliary features are dropped from german, svmguide3, magic04 and a8a datasets respectively (see Table 4, page 12) to make sure the same amount of overall data is unavailable.
>
> 4. Overall, I suggest the authors better define the problem/scope of this paper and try to prevent over generalizing the proposed approach.
> - Thank you for pointing this out. We believe that we have narrowed down the scope of what is not considered haphazard inputs (see the last paragraph "Haphazard Inputs and Aux-Drop vs Other Works" in section 3.4, Page 9).

---

> > ### Comment · Reviewer_MXSX · 2023-05-16
> > **Thanks for the response; follow-on question on the assumption of base-features**
> >
> > I’d like to thanks the authors for addressing my questions in the rebuttal and better define the problem of Haphazard Inputs.
> >
> > In the revised manuscript, I'm wondering would it be possible to remove the assumption "there is at least one feature always available (called base feature)"? I think it could be a more elegant and self-consistent problem if any feature can be missing, rather than the assumptions of a set of base feature.

---

> > > ### Author Response · Authors · 2023-05-18
> > > **Response on assumption of base-features**
> > >
> > > We agree with you that without the assumption of a set of base features, the problem would be more elegant and self-consistent. But in the case of Aux-Drop, it is a limitation and requirement of the model to assume at least one base feature. However, the Aux-Drop model can be adapted in multiple ways to remove the assumption of base features as follows:
> > > - The naive way to handle this would be to just assume one auxiliary feature as a base feature and impute it. This would induce some bias but still, it would be for just one feature.
> > > - Choosing the position of AuxLayer as the first layer, the Aux-Drop model can be adapted to handle haphazard inputs without any base feature.
> > >
> > > But we attempt to propose a generalized concept of Aux-Drop where the position of AuxLayer is a hyperparameter and can be chosen based on the dataset and application. Hence, the assumption of the base feature is adding a better value proposition.

---

### Review · Reviewer_hDLm · 2023-04-06

**Summary Of Contributions:**

This paper studies online learning which produces streaming data. The authors claim that streaming data is haphazard in nature, which means data contains missing features, features becoming obsolete in time, the appearance of new features at later points in time, and a lack of clarity on the total number of input features. The existing haphazard inputs make it hard to build a learnable system. To tackle this challenge, an auxiliary dropout regularization strategy named Aux-Drop is proposed to help in better learning for scenarios where certain features disappear in time or when new features are to be modeled. Experimental results show the effectiveness of the proposed method.

**Audience:**

Yes

**Broader Impact Concerns:**

No.

**Claims And Evidence:**

Yes

**Requested Changes:**

1.Technical contributions are limited. Now the contribution in the proposed method is engineering contribution rather than technical contribution in the perspective of machine learning.

2.How the proposed method can handle the haphazard inputs is not very clear.

3.For further improving this work, I would suggest clarifying concepts before diving into the technical section. For example, I do not think the concept of Haphazard Inputs  is well-established. Also, this concept can not be found in any paper. It would be better if the authors can convince me this is a well-established concepts in machine learning community.

**Strengths And Weaknesses:**

Strengths:

1.Online learning is a real problem.

2.This paper is well-written.

Weaknesses:

1.Technical contributions are limited. Now the contribution in the proposed method is engineering contribution rather than technical contribution in the perspective of machine learning.

2.How the proposed method can handle the haphazard inputs is not very clear.

3.For further improving this work, I would suggest clarifying concepts before diving into the technical section. For example, I do not think the concept of Haphazard Inputs  is well-established. Also, this concept can not be found in any paper. It would be better if the authors can convince me this is a well-established concepts in machine learning community.

---

> ### Author Response · Authors · 2023-04-20
>
> We thank the reviewer for their comments and try to resolve them as much as possible.
>
> 1. Technical contributions are limited. Now the contribution in the proposed method is engineering contribution rather than technical contribution from the perspective of machine learning.
> - It looks like our writing has some drawbacks which made you perceive less technical contribution. But since then, the paper is rewritten (changes are shown in blue color) and we hope now that technical contribution is evident. The technical contribution is the way in which auxiliary dropout is applied in the AuxLayer. Yes, it is a very small contribution in the field of dropout but when applied to a deep-learning-based online learning framework, it handles the problem of haphazard inputs seamlessly.
>
> 2. How the proposed method can handle the haphazard inputs is not very clear.
> - We now include a subsection "Handling Haphazard Inputs" (see section 3.3, page number 6, 7 and 8) which shows how Aux-Drop handles each characteristic of the haphazard inputs.
>
> 3. For further improving this work, I would suggest clarifying concepts before diving into the technical section. For example, I do not think the concept of Haphazard Inputs is well-established. Also, this concept can not be found in any paper. It would be better if the authors can convince me this is a well-established concept in the machine learning community.
> - We thank the reviewer for pointing this out. The introduction is rewritten and we think now it addresses this point. The first paragraph of the introduction (discusses the characteristics of haphazard inputs and the second paragraph refers to the previous works where haphazard inputs are introduced. Please note that in the previous work, the characterization of the problem discussed here is not defined properly, hence we make an attempt to introduce the problem (haphazard inputs) in a more formal way in this paper.

---

### Comment · Action_Editors · 2023-05-05
**Official recommendation overdue**

Dear reviewers,

Your official recommendation is overdue. Could you please submit it at your earliest convenience? Thank you!

AE

---

### Author Response · Authors · 2023-05-27
**Token of thanks**

We thank the action editor for taking out time and conducting a thorough review of the paper within a short span of time. We would also like to thank all the reviewers for their comments. It has improved the manuscript tremendously. Thank you everyone.

---

### Decision · Action_Editors · 2023-05-22

**Recommendation:** Accept with minor revision

**Comment:**

The paper studied online learning with "haphazard inputs" that is a new concept defined as "missing features, features becoming obsolete in time, the appearance of new features at later points in time and a lack of clarity on the total number of input features". A dropout-inspired method called Aux-Drop (an auxiliary dropout regularization strategy) was proposed to solve the problem. In the beginning, the reviewers questioned that the contributions were quite engineering but not scientific and the details provided in the paper and the code were not enough to reproduce the results. During the rebuttal period, the authors did a particularly good job --- several parts have been rewritten and the concerns have been addressed. In the end, all reviewers agreed to accept the paper for publication. Moreover, Reviewer 7CMD commented that "this is an interesting paper that meets the technical soundness and reproducibility bars" and recommended the Reproducibility Certification, and I am happy to make the same recommendation. In the camera-ready version, please make sure to incorporate the discussion from the reply "response on assumption of base-features".

**Audience:**

Yes.

**Claims And Evidence:**

Yes, after the rebuttal when the introduction has been rewritten, the claims are well supported.